# DATA GENERATION BASED ON DIFFUSION GEOMETRY

**Ofir Lindenbaum, Jay S. Stanley III, Guy Wolf & Smita Krishnaswamy**
Yale University, New Haven, CT, USA

## ABSTRACT

Many generative models attempt to replicate the density of their input data. However, this approach is often undesirable, since data density is highly affected by sampling biases, noise, and artifacts. We propose a method called SUGAR (Synthesis Using Geometrically Aligned Random-walks) that uses a diffusion process to learn a manifold geometry from the data. Then, it generates new points evenly along the manifold by pulling randomly generated points into its intrinsic structure using a diffusion kernel. SUGAR equalizes the density along the manifold by selectively generating points in sparse areas of the manifold. We demonstrate how the approach corrects sampling biases and artifacts, while also revealing intrinsic patterns (e.g. progression) and relations in the data.

## 1 INTRODUCTION

Many types of data analysis suffer from what is known as "imbalanced data" or biased sampling of data from a system. Statistical methods such as mutual information are highly weighted by density and thus can mis-quantify the strength of dependencies with data whose density is concentrated in a particular region of the relationship. For example, in immunology, if the activity of a t-cell stimulatory molecule and its respondent inflammatory cytokines is mostly observed in an off state, statistical measures will be biased towards negligible dependency, despite the true positive relationship between t-cell stimulation and inflammation.

In principle component analysis, major components (eigenvectors of the covariance matrix) can span data points of a particular type if they are oversampled. For instance, if healthy individuals form 99 percent of a sample then PC components may span variation in healthy individuals rather than the gradient between healthy and sick patients. In supervised learning, some classifiers tend to bias towards the classes with highest density He & Garcia (2009); López et al. (2013); Hensman & Masko (2015). Most other generative models attempt to learn and replicate the density of the data, which is not only intractable in high dimensions but also exacerbates this problem. Examples for such methods include Gaussian Mixture Models (GMM) Rasmussen (2000), variational Bayesian methods Bernardo et al. (2003), kernel density estimates Scott (2008).

We assume that the sampled data lie on low-dimensional manifolds. Based on this, we propose a new type of generation method termed SUGAR (Synthesis Using Geometrically Aligned Random-walks) that learns the underlying manifold geometry of the data. Under-sampled regions within the manifold geometry can be regenerated using this structure. We learn the manifold by using a diffusion operator or a kernel that requires only the computation of pairwise affinities between data points. Then, we generate new points randomly around existing points. Finally, we apply a weighted transition kernel to pull the new points towards the structure of the manifold, especially in sparse areas.

## 2 PROBLEM FORMULATION

Let $\mathcal{M}$ be a $d$ dimensional manifold that lies in a higher dimensional space $\mathbb{R}^D$, with $d < D$, and let $\boldsymbol{X} \subseteq \mathcal{M}$ be a dataset of $N = |\boldsymbol{X}|$ data points, denoted $\boldsymbol{x}_1, \ldots, \boldsymbol{x}_N$, sampled from the manifold. In this paper, we propose an approach that uses the samples in $\boldsymbol{X}$ in order to capture the manifold geometry and generate new data points from the manifold. In particular, we focus on the case where the points in $\boldsymbol{X}$ are unevenly sampled from $\mathcal{M}$, and aim to generate a set of $M$ new data points $\boldsymbol{Y} = \{\boldsymbol{y}_1, ..., \boldsymbol{y}_M\} \subseteq \mathbb{R}^D$ such that 1. the new points $\boldsymbol{Y}$ approximately lie on the

manifold $\mathcal{M}$, and 2. the distribution of points in the combined dataset $\boldsymbol{Z} \triangleq \boldsymbol{X} \cup \boldsymbol{Y}$ is uniform. Our proposed approach is based on using intrinsic diffusion process over the manifold to define a diffusion geometry that robustly captures the manifold geometry even from $\boldsymbol{X}$. Then, we use this diffusion process to generate new data points that follow the manifold geometry while adjusting their intrinsic distribution, as explained in Section 3.

## 3 METHOD

In this section, we detail the proposed method named SUGAR: Synthesis Using Geometrically Aligned Random-walks. SUGAR initializes by synthesizing new points around sparse areas of the manifold to create a new set of points $\boldsymbol{Y}_0$. SUGAR is summarized in the following steps:

- Construct a kernel that captures local neighborhoods in the data
  $\mathcal{K}(\boldsymbol{x}_i, \boldsymbol{x}_j) \triangleq K_{i,j} = exp\left(-\frac{||\boldsymbol{x}_i - \boldsymbol{x}_j||^2}{2\sigma^2}\right), i, j = 1, ..., N$ where $\sigma$ is a user-configurable parameter that controls the neighborhood sizes. This kernel captures the diffusion geometry, and was utilized in DM Coifman & Lafon (2006) for dimensionality reduction.

- Compute the degree of the kernel, defined as $\hat{d}(i) = \sum_j exp\left(-\frac{||\boldsymbol{x}_i - \boldsymbol{x}_j||^2}{2\sigma^2}\right), i = 1, ..., N$. The degree value $\hat{d}(i)$ at each point indicates the amount of connectivity the point has to its neighbors.

- Use the degree $\hat{d}(i)$ to define sparsity of each point as $\hat{s}(i) \triangleq \frac{1}{\hat{d}(i)}, i = 1, ..., N$. Clearly $\hat{s}(i) \geq 0, i = 1, ..., N$, as the degree $\hat{d}(i)$ at each point is non-negative.

- Define the generation level $\hat{\ell}(i)$ at each point $i = 1, ..., N$, by
  $\hat{\ell}(i) = \lfloor \det(\boldsymbol{\Sigma}_i^{-1} + \frac{\boldsymbol{I}}{2\sigma^2})^{0.5} \det(\boldsymbol{\Sigma}_i)^{0.5} [\max(\hat{d}(\cdot)) - \hat{d}(i)] \rfloor$, where $\hat{d}(i)$ is the degree value at point $\boldsymbol{x}_i$, $\sigma^2$ is the bandwidth of the kernel $\boldsymbol{K}$ and $\boldsymbol{\Sigma}_i$ is the covariance of the Gaussian designed for generating new points (explained in the following step). By generating $\hat{\ell}(i), i = 1, ..., N$ new points, the expectation value of the degree $\hat{d}(i)$ at each point $\boldsymbol{x}_i \in \mathcal{M} \backslash \partial \mathcal{M}$ is constant.

- Draw $\hat{\ell}(i)$ new points for each $i = 1, ..., N$ from a Gaussian distribution designed to maintain the local structure. Each Gaussian $\mathcal{N}(\boldsymbol{x}_i, \boldsymbol{\Sigma}_i)$ is centered around an existing point $\boldsymbol{x}_i$ and has a local covariance structure $\boldsymbol{\Sigma}_i$. The local covariance $\boldsymbol{\Sigma}_i$ is the sample covariance based on $k$ nearest neighbors surrounding $\boldsymbol{x}_i$. Thus, we elaborate the local structure of the manifold to generate points in the meaningful directions. The set of new points is denoted by $\boldsymbol{y}_i \in \boldsymbol{Y}_0, i = 1, ..., M$.

- Compute the sparsity based MGC kernel is defined as
  $\hat{\mathcal{K}}(\boldsymbol{y}_i, \boldsymbol{y}_j) = \sum_\ell \mathcal{K}(\boldsymbol{y}_i, \boldsymbol{x}_\ell) \mathcal{K}(\boldsymbol{x}_\ell, \boldsymbol{y}_j) \hat{s}(\ell), i, j = 1, ..., M$. Normalize $\hat{\boldsymbol{P}} = \hat{\boldsymbol{D}}^{-1} \hat{\boldsymbol{K}}$. using a diagonal matrix $\hat{\boldsymbol{D}}$, such that $\hat{D}_{i,i} = \sum_j \hat{K}_{i,j}$.

- Apply the operator $\hat{\boldsymbol{P}}$ at time instant $t$ to the new generated points. The diffused points are defined as $\boldsymbol{Y} = \hat{\boldsymbol{P}}^t \cdot \boldsymbol{Y}_0$. The operator locally averages the points based on the neighbors in $\boldsymbol{X} \subset \mathcal{M}$ and the sparsity measure. The number of steps $t$ required can be set manually or using the Von Neumann Entropy as was suggested in Moon et al. (2017).

## 4 ARTIFICIAL MANIFOLD

Here we evaluate how SUGAR captures a manifold structure and generates points for density equalization. We sample a hundred points from a circle such that the highest density is at the origin ($\theta = 0$) and the density decreases away from it. Figure 1(a) shows each point $\boldsymbol{x}_i$ colored by its degree $\hat{d}(i)$ (as defined in Eq. 3). Figure 1(e) shows new points generated based on $\hat{\ell}(i)$ around each original point.

We apply SUGAR and estimate the CDF of one coordinate from $\boldsymbol{X}$ and $\boldsymbol{Z}$. The resulting CDF shown in blue is approximately a standard uniform CDF as shown in red in Fig. 1(f). The variance of the degree $\hat{d}(\cdot)$ drops from 0.45 to 0.03, yet another indication of the improved density of points.

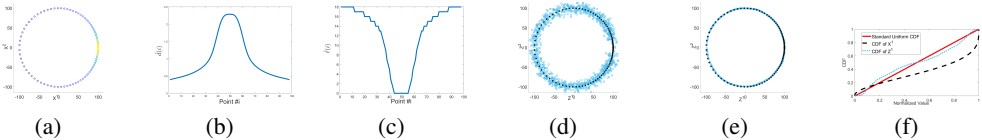

(a)        (b)        (c)        (d)        (e)        (f)

Figure 1: Density equalization demonstrated on a circle shaped manifold. (a) Non-uniform samples of $X$. (b) The degree value $\hat{d}(i)$. (c) The number of generated points $\hat{\ell}(i)$. (d) $X$ (black asterisks) and generated points $Y_0$ (blue circles). (e) $Z$, original points $X$ (black asterisks) and set of new points $Y$ (blue circles). (f) An estimation of the CDF before (black asterisks) and after (blue circles) applying the SUGAR.

## 5 BIOLOGICAL MANIFOLDS

In this section, we apply SUGAR to a biological dataset. In this data, Velten et al. (2017) present a model of cellular development in which a central reservoir of stem cells gives rise to unique, continuous trajectories, each leading to functionally distinct mature cells. This branching geometry thus lends itself to exploration using manifold learning. However, the size of this data is a key obstacle: only 1029 cells are measured in one individual. The data is thus generally sparse, leading to rare populations and branch discontinuities in cellular development due to undersampling. We sought to repair this sparsity using SUGAR.

We first visualized this data using the unsupervised dimensionality reduction technique PHATE (figure 2(a), colored by density). This embedding captured cellular development trajectories (figure 2(b)) similar to the supervised approach employed in Velten et al. (2017). SUGAR was then applied to the original data and this embedding to generate 1919 new points (for a total of 2948 cells).

Despite nearly tripling the size of the data, the SUGAR-generated points were faithful to canonical biology and data geometry. Transition populations spanning biological time between mature cells and the stem cell reservoir are enriched by SUGAR, augmenting phase transitions that were once sparse. Two examples of this restoration are present in this data. First, the granulocyte (neutrophils (N) and Eosinophil/Basophil/Mast Cells (EBM); Murphy & Weaver 2016)) transition (top right branch of figure 2(a) labeled in 2(b)) is discontinuous between the dense neutrophil/stem-cell body and the dense EBM island. SUGAR repaired this discontinuity, illustrated by the EBM gene profile (Velten et al. (2017)) shown by the color in figure 2(c). Second, the original dataset contains a large gap between a dense plane of B-primed cells and a sparse island of more mature B cells (rightmost branch in figure 2(a), labeled in 2(b)). SUGAR recovered this trajectory, evidenced by transitional up-regulation of the B cell maturation marker *CD19* (figure 2(d)). Furthermore, SUGAR recovered an inverse relationship between *CD19* and the cell immaturity marker *HOXA3* (figures 2(e), 2(f))These examples illustrate the ability of SUGAR to recover rare or difficult to capture populations while maintaining data geometry.

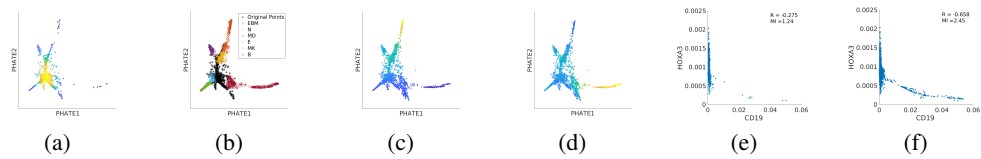

(a)        (b)        (c)        (d)        (e)        (f)

Figure 2: SUGAR recovers branching trajectories in hematopoeisis. (a) PHATE plot of the original Velten et al. (2017) data colored by the degree $\hat{d}(i)$. (b) PHATE plot of Velten et al. (2017) (black asterisks) data with SUGAR generated points $Y$ (circles) colored by genetic module profile. EBM: Eosinophil/Basophil/Mast Cells; N: Neutrophils; MD: Monocytes/Dendritic Cells; E: Erythroid; MK: Megakaryocyte; B: B cell. (c) EBM Module Expression. (d) *CD19* count after SUGAR. *CD19* is found in maturing B cells Murphy & Weaver (2016) (e) Relationship between *HOXA3* and *CD19* before SUGAR. The *HOXA3/HOXB6* module marks non-committed stem cells Velten et al. (2017) (f) Relationship between *HOXA3* and *CD19* after SUGAR.
.

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
