# OpenReview forum: "Data Generation Based on Diffusion Geometry"
_ICLR.cc/2018/Workshop — Reject_

### Official Review · AnonReviewer2 · 2018-02-21
**I don't quite understand the problem**

**Rating:** 5
**Confidence:** 2

**Review:**

The paper consider scenarios where parts of the feature space are drastically under-sampled, e.g. as in cases where one class is over-represented, while another is under-represented. The paper argues standard generative models will fare poorly in such scenarios as they model the PDF of the data, whereas the "right thing" is to model the underlying data manifold. The paper then provides an algorithm for approximately uniformly sampling on this data manifold. This is then argued to be a better generative model, than one that captures the PDF of the data.

I'll confess that I don't really understand the problem that is being solved here. To me, a generative model is nothing more than a density estimator. It seems the authors think of generative models in a different way, which I don't really follow. What is it exactly the proposed model is supposed to be used for? My inability to answer this question is my main concern with the paper (but I acknowledge that I most likely missed something important in the paper)

A few questions:
1) It seems the intrinsic manifold dimensionality (d) is never actually used -- is this correct?

2) If we consider the case of a linear manifold, then would the proposed method (essentially) reduce to doing PCA on the data, and then uniformly sampling in the latent space within the convex hull of the latent variables? If so, why is that something you want?

3) The manifold seems to be captured by the diffusion kernel if I understand correctly. If we sample uniformly over (a region of) the manifold, what is then the PDF from which we draw samples? Given the uniform distribution of the manifold, I would expect that you could actually derive this PDF -- I suspect it would be a variant of a kernel density estimator.

A side note:
The idea of capturing the shape of the manifold by a local covariance was recently explored by Arvanitidis et al (NIPS 2016), who then fit a Riemannian normal distribution to the data under the local metric induced by the local covariance. Samples from this distribution appear to be fairly similar to a uniform distribution over the data support. It seems like there's a conceptual link.

---

### Official Review · AnonReviewer1 · 2018-03-04
**Authors proposed a method called SUGAR that uses a diffusion process to learn a manifold geometry from the data, and it generates new points evenly along the manifold by pilling randomly generated points into its intrinsic structure using a diffusion kernel.**

**Rating:** 5
**Confidence:** 4

**Review:**

Authors proposed a method called SUGAR that uses a diffusion process to learn a manifold geometry from the data, and it generates new points evenly along the manifold by pilling randomly generated points into its intrinsic structure using a diffusion kernel.

The studied problem is interesting when the intrinsic structure is unknown. However, the fundamental assumption of SUGAR is that the true diffusion manifold is obtained by diffusion process. Some concerns for this assumption are listed as follows:

First, authors claimed that SUGAR could be helpful for the data (In Abstract and Introduction), the density of which is highly affected by sampling biases, noise and artifacts. It seems that the diffusion process is the key to capture the true manifold by assumption, not SUGAR itself. If a sparse region is really captured by diffusion process, does SUGAR still generate more points to fill up this sparse region? It is still a question needed to answer: how to distinguish the true or false sparse regions to generate points.

Second, if the true intrinsic manifold structure is known, it might be less important/difficult to generate new points evenly along the manifold. For the visualization purpose, it might be nice to show low-dimensional points evenly along the manifold using generated points. However, it still comes from perceptual understanding since the disconnection or large gap is hard to measure in the experiments. Existing methods can automatically represent the manifold by graph structures. In this case, generating more points is not more representative than graph structures (or trajectories).

Third, it is possible that the diffusion process incorrectly recover the manifold structure. How SUGAR is affected? It might be interesting to see the generated data interacts with the learned manifold structure iteratively to obtain a better manifold structure. More results should be conducted by comparing with existing methods to capture the biological manifolds.

In Section 3, it is bit confusing about the process and symbols.

-  Since X is a d-dimensional sample, how to get it from observations data in a D-dimensional space. I guess that PHATE is used in Section 5, but which method is used in Section 4. Y_0 is unclear.

- In Section 2, Y is D-dimensional. How to calculate Y = P^t Y_0 and visualize Y in a 2-d space as shown in Figure 2.

- No Eq. 3

- No colorbar to show the density in Figure 2(a), for which color represents high/low density.

---

### Official Review · AnonReviewer3 · 2018-03-05
**Too weak arguments and theoretical validation**

**Rating:** 3
**Confidence:** 3

**Review:**

The paper proposes a method for balancing imbalanced datasets by sampling new data points in the low-dimensional submanifold of the data space the samples are assume belonging to. It uses a diffusion kernel to randomly generate new points in sparse areas of the manifold. The method is evaluted on synthethic data and biological datasets.

The method appears as a sequence of steps that in combination is denoted SUGAR. In essence, a Gaussian kernel is used to define locality and thereby capture the geometry of the existing data point. New points are then sampled to even out the degree of sparsity. The points are then mapped to the manifold.

The paper lacks in its theoretical foundation. It is not argued why the SUGAR steps achieves the aim of producing uniform data. From a manifold learning perspective, the sequence of steps are natural but arguments for the validity of the method are not present. Sampling from a given measure on a manifold is not trivial, and a new proposed method needs to present arguments for its validity. At this stage, it is now clear if the method achieves its aim of producing uniform samples.

The approach can potentially have some merit in augmenting datasets that span submanifolds of the data space but I do not believe it currently is ready to be published.

---

### Decision · Program_Chairs · 2018-03-20
**ICLR 2018 Workshop Acceptance Decision**

**Decision:**

Reject

**Comment:**

Based on the reviews, this paper has not been accepted for presentation at the ICLR workshop. However, the conversation and updates can continue to appear here on OpenReview.